# SIMPLE-TTS: END-TO-END TEXT-TO-SPEECH SYNTHESIS WITH LATENT DIFFUSION

## ABSTRACT

We propose an end-to-end text-to-speech (TTS) latent diffusion model as a simpler alternative to more complicated pipelined approaches for TTS synthesis. In particular, we show that one can adapt a recently proposed text-to-image diffusion architecture, U-ViT, as an excellent backbone for audio generation. We identify and explain the changes required for this adaptation and demonstrate that latent diffusion is an effective approach for end-to-end speech synthesis, without the need for phonemizers, forced aligners, or complex multi-stage pipelines. Despite its simplicity, our proposed approach, Simple-TTS, outperforms more complex models that rely on explicit alignment components and significantly outperforms the best open-source multi-speaker TTS system. We will open-source Simple-TTS upon acceptance, making it the strongest system publicly available to the community. Due to its straight-forward design, we expect that Simple-TTS can easily be adapted to many diverse TTS settings — opening the stage to repeat the success of Stable Diffusion in computer vision, in audio generation.

## 1 INTRODUCTION

The capabilities of generative models have advanced rapidly across a range of modalities. This progress has been driven by the emergence of simple, end-to-end solutions that require minimal supervision and are highly scalable. However, the approaches differ across data modalities. For discrete domains such as language, the most effective models have been autoregressive transformers (Brown et al., 2020). For continuous domains, such as images and videos, diffusion models have lately led to impressive improvements (Ho et al., 2020; 2022b). Both lines of approaches are currently being explored for audio generation (Wang et al., 2023; Le et al., 2023).

Audio generation presents challenges for autoregressive methods due to the large lengths of audio waveforms at common sampling rates (e.g. 16kHz). To handle such long inputs, recent autoregressive approaches utilize neural models to tokenize the waveform into a shorter sequence of audio tokens (Borsos et al., 2023). Vall-E, for instance, is an autoregressive TTS system that utilizes a vector-quantized autoencoder (van den Oord et al., 2017) to tokenize raw waveforms to 600 audio tokens per second (Wang et al., 2023). A 10 second clip of audio is therefore represented with 6,000 instead of 160,000 discrete values (at a 16kHz sampling rate). Even with such audio tokenization, autoregressive approaches such as AudioLM and Vall-E still utilize cascades of multiple models that first generate coarse acoustic features and then finer details (Borsos et al., 2023).

Diffusion models (Sohl-Dickstein et al., 2015; Ho et al., 2020), on the other hand, are a natural fit for end-to-end text-to-speech generation. Their iterative sampling procedure enables a single model to generate audio in a coarse-to-fine manner, while parallel generation improves their scalability to long sequences. However, so far text-to-speech diffusion models have been rather complex (Shen et al., 2023; Le et al., 2023). For example, they require explicit phoneme duration annotations to perform speech synthesis. For training, this means that such systems are limited to datasets with frame-level phonetic transcripts. For generation, obtaining this alignment requires training external duration prediction models which introduces potential performance bottlenecks and increases system complexity. Further, they typically require multi-stage generation pipelines, specialized tools such as phonemizers or forced aligners, and additional trainable components.

Recent advances in neural audio autoencoders enable high-fidelity compression of raw audio waveforms (Défossez et al., 2022). Building upon this progress, we explore latent diffusion (Rombach

et al., 2021) as a promising paradigm for generative modeling of audio. We use the publicly available EnCodec (Défossez et al., 2022) to compress waveforms to a sequence of 75 continuous vectors per second. For our diffusion network, we adapt the U-Vision Transformer (U-ViT), a recently proposed image diffusion architecture, to operate on 1D sequence data instead of images. Our proposed architecture, termed the U-Audio Transformer (U-AT), consists of a 1-dimensional U-Net and a transformer backbone. We apply the U-Net encoder to downsample the lengthy audio features and then process the downsampled sequence with the transformer backbone. We upsample the features back to the original sequence length with the U-Net decoder for the final prediction. This design enables us to efficiently apply a strong transformer backbone to long sequence data. We demonstrate that U-AT is an excellent backbone for audio generation.

Effective speech synthesis relies on fine-grained language characteristics like spelling, which are discarded by subword tokenization methods such as byte-pair encoding (BPE) (Sennrich et al., 2016). We therefore condition our network on representations from the byte-level language model ByT5 (Xue et al., 2022) to enable TTS generation. ByT5 is pre-trained with the T5 span corruption objective (Raffel et al., 2020) upon raw UTF-8 byte sequences. As a result, ByT5 is character-aware, unlike most popular language models. Because words with similar spelling often have similar pronunciations, character-aware representations can improve generalization. BPE models, on the other hand, often represent similar words (e.g. chair and chairs) with independent tokens. We empirically validate that embedding the transcript with ByT5 instead of T5, which is widely used by text-to-image models (Saharia et al., 2022a), is critical for generating coherent speech.

Balaji et al. (2022) observed that the behavior of text-to-image diffusion models evolves over the course of the diffusion process. At high noise levels with high uncertainty, the network relies heavily upon the text description. However, as the level of noise decreases, the model increasingly relies upon the visual features and ignores the description. To increase our model's utilization of the transcript, we modify the diffusion noise schedule to dedicate more training time to high noise levels where the structure of the speech (duration, word placement, etc.) is uncertain and the transcript is most useful. This significantly improves the alignment of the synthesized speech with the transcript.

Our latent diffusion model, Simple-TTS, is the first diffusion model capable of end-to-end TTS synthesis. Simple-TTS outperforms strong open-source baselines with explicit alignment components and is conceptually simpler than existing TTS diffusion models. We compare the system complexity of recent diffusion TTS models in Table 1. Given a transcript, Simple-TTS is the only diffusion model capable of generating speech without additional conditioning information. To evaluate the intelligibility of the synthesized speech, we transcribe it using a pre-trained speech recognition model. We achieve a word error rate (WER) of 2.4%, outperforming recent TTS systems and nearly matching the reference WER of 2.2% achieved when transcribing natural human speech.

We also extend our system to speaker-prompted TTS synthesis. Given a speaker prompt, Simple-TTS generates speech that maintains the voice and style of the prompt. In this setting, our system produces more intelligible speech than the state-of-the-art autoregressive system Vall-E (WER of 3.4% vs. 5.9%). Simple-TTS is also more effective at matching the characteristics of the prompt than the strongest open-source system YourTTS (speaker similarity of 0.514 vs. 0.337). We conduct a human evaluation and find that Simple-TTS achieves statistically significant improvements in human judgements of quality and prompt similarity compared to YourTTS. We present uncurated audio samples from Simple-TTS in the supplemental material to showcase the effectiveness of our system. Upon acceptance, we will open-source Simple-TTS, making it the strongest system publicly available to the community. Our work demonstrates the viability of latent diffusion for text-to-speech synthesis and paves the way for further scaling and improvements of generative speech systems.

## 2 RELATED WORK

The most related systems are the proprietary diffusion TTS models, NaturalSpeech2 (Shen et al., 2023) and the concurrent VoiceBox (Le et al., 2023). We contrast these systems with Simple-TTS in Table 1. Both systems require phonemizers and a forced aligner to produce frame-level phonetic transcripts for training, which can be error-prone. Many phonemizers, for instance, operate on the word-level and ignore neighboring words when predicting the pronunciation (McAuliffe et al., 2017). Such approaches fail to handle situations where the pronunciation is context-dependent. We

| Method | Prerequisites | Trained Components | Generation Inputs |
|---|---|---|---|
| NaturalSpeech2 (Shen et al., 2023) | • Frame-level phonetic transcripts
• Phonemizer
• Audio autoencoder | • Phoneme duration predictor
• Pitch predictor
• Phoneme encoder
• Speech prompt encoder
• Diffusion network | • Transcript
• Phoneme durations
• Framewise pitch
• Speech prompt |
| VoiceBox (Le et al., 2023) | • Frame-level phonetic transcripts
• Phonemizer
• HiFi-GAN vocoder | • Phoneme duration predictor
• Diffusion network | • Transcript
• Phoneme durations |
| Simple-TTS (Ours) | • Text transcripts
• Pre-trained language model
• Audio autoencoder | • Diffusion network | • Transcript |

Table 1: Comparison of our approach, Simple-TTS, with prior TTS diffusion methods.

demonstrate that the contextual byte representations from a pre-trained ByT5 model can replace phoneme representations as a strong conditioning signal for TTS synthesis.

Similar to Simple-TTS, NaturalSpeech2 learns a diffusion model in the latent space of an audio autoencoder, but they develop their own autoencoder. We demonstrate that publicly available systems, such as EnCodec, are of sufficient quality to learn an effective latent diffusion model. VoiceBox generates Mel spectrogram features, which are then decoded to raw waveforms with a HiFi-GAN vocoder (Kong et al., 2020) trained in addition to their diffusion model.

Both systems require phoneme duration annotations for generation. Using either method therefore requires training an external duration prediction model. NaturalSpeech2 trains a regression model for phoneme duration prediction and can therefore only generate speech of a single duration for any given prompt-text pair. Our system is capable of synthesizing diverse speech across the distribution of natural durations. VoiceBox overcomes the limitation of regression-based duration models by learning an additional diffusion model for phoneme duration prediction. The duration diffusion model can perform one-to-many predictions, but increases the complexity of the overall system. NaturalSpeech2 also requires framewise pitch annotations during training and generation which requires training an additional pitch prediction model. The system also requires a speech prompt for generation and, unlike Simple-TTS, is incapable of text-only generation using just the transcript.

Diffusion models have also been developed for audio applications beyond text-to-speech synthesis. For instance, diffusion models have been leveraged to develop high-quality neural vocoders, which synthesize audio waveforms conditioned on the mel spectrogram of the original audio Chen et al. (2020); Kong et al. (2021). Diffusion models have also been developed for the distinct setting of text-to-*audio* generation, as opposed to text-to-*speech* generation. This involves generating clips of audio given some text description of the content (e.g. a clip of "a dog barking") (Liu et al., 2023; Huang et al., 2023). However, such text-to-audio models are not capable of generating coherent speech, which requires fine-grained phonetic, in addition to semantic, language understanding. Simple-TTS, in contrast, leverages representations from a byte-level language model to capture nuanced phonetic properties critical for intelligible speech generation.

## 3 BACKGROUND

Diffusion models (Sohl-Dickstein et al., 2015; Ho et al., 2020; Kingma et al., 2021) are latent variable models with latents $\mathbf{z} = \{\mathbf{z}_t | t \in [0, 1]\}$ given by a forward diffusion process $q(\mathbf{z}|\mathbf{x})$, which defines a gradual transition from the data distribution, $\mathbf{x} \sim p(\mathbf{x})$, to a Gaussian distribution. The Markovian forward process iteratively adds Gaussian noise to the data over time and satisfies

$$q(\mathbf{z}_t|\mathbf{x}) = \mathcal{N}(\mathbf{z}_t; \alpha_t\mathbf{x}, (1 - \alpha_t^2)\mathbf{I}), \quad q(\mathbf{z}_t|\mathbf{z}_s) = \mathcal{N}(\mathbf{z}_t; \alpha_{t|s}\mathbf{z}_s, (1 - \alpha_{t|s}^2)\mathbf{I})$$

where $\alpha_{t|s} = \alpha_t/\alpha_s$ and $0 \le s < t \le 1$. The noise schedule, determined by $\alpha_t \in [0, 1]$, monotonically decreases the signal-to-noise ratio (SNR), $\lambda_t = \frac{\alpha_t^2}{1-\alpha_t^2}$ as a function of the time, $t$, such that

the final latent becomes approximately Gaussian, $q(\mathbf{z}_1) \approx \mathcal{N}(\mathbf{0}, \mathbf{I})$. The forward process therefore defines a transition from the data distribution to a Gaussian distribution.

Diffusion models define a generative process by inverting the forward process. This specifies a transition from Gaussian noise, which can be sampled analytically, to the unknown data distribution. Inverting this process can be reduced to learning a *denoising network*, $\hat{\mathbf{x}}_\theta(\mathbf{z}_t, t, \mathbf{c}) \approx \mathbf{x}$, that reconstructs the clean data given some noisy latent, the time, and (optionally) some conditioning information, $\mathbf{c}$, about the data. The conditioning information could be a textual description of an image (Saharia et al., 2022a) or, in our case, a textual transcription of some speech. This denoising network is trained with a regression objective

$$\mathcal{L}(\theta) = \mathbb{E}_{t,\mathbf{x},\epsilon}[w(\lambda_t) \|\hat{\mathbf{x}}_\theta(\mathbf{z}_t, t, \mathbf{c}) - \mathbf{x}\|_2^2]$$

with some time-dependent weighting, $w(\lambda_t)$, that is set empirically to emphasize noise levels that are important for downstream perceptual quality (Ho et al., 2020; Nichol & Dhariwal, 2021).

This loss function is the weighted variational lower bound of the log likelihood of the data under the forward diffusion process (Sohl-Dickstein et al., 2015; Ho et al., 2020; Kingma et al., 2021). In practice, the denoising network is often parameterized as a noise prediction network (Ho et al., 2020) or a velocity prediction network (Salimans & Ho, 2022) where the velocity, $\mathbf{v}$, is defined as $\mathbf{v} = \sqrt{\alpha_t}\epsilon - \sqrt{1 - \alpha_t}\mathbf{x}$. These parameterizations can be interpreted as different weighting functions, $\lambda_t$, for the regression objective (Salimans & Ho, 2022). We adopt the $\mathbf{v}$-parameterization throughout this work. We use the standard DDPM sampler with 250 sampling steps for generation.

## 4 SIMPLE-TTS

**Latent Audio Diffusion.** We utilize the publicly available audio autoencoder, EnCodec (Défossez et al., 2022), to map waveforms to a sequence of 75 latent vectors per second. EnCodec, like other audio autoencoders (Zeghidour et al., 2021), applies residual vector quantization to map each continuous vector to a variable number of discrete tokens that capture increasingly fine details. The number of quantizers can be adjusted to trade off compression rates and quality. This quantization produces lengthy representations that are challenging to model. When using all 32 quantizers, the 75 continuous representations are quantized to 2,400 discrete codes. Prior autoregressive approaches reduce the number of quantizers, which degrades audio quality, and develop multi-stage pipelines with specialized models for generating tokens from early and late quantizers.

Diffusion models, on the other hand, can directly generate continuous representations, avoiding the need for discrete tokenization. We take advantage of this by training our model to produce the 128-dimensional continuous embeddings from the EnCodec encoder, before vector quantization. This decision significantly reduces the sequence length: with all quantizers, a 10 second clip consists of 750 latents rather than 24,000 (a 32x reduction). The continuous latents generated during inference can then be quantized and decoded by EnCodec to recover the raw audio waveform.

**U-Audio Transformer (U-AT).**

For our diffusion network, we adapt the U-Vision Transformer (U-ViT), a recently proposed image diffusion architecture, to operate on 1D sequence data instead of images. Because we focus on audio generation, we refer to our model as the U-Audio Transformer (U-AT) and present an overview in Figure 1. The U-AT consists of a 1D U-Net and a transformer backbone. We apply the U-Net encoder to downsample the lengthy audio features and then process the downsampled sequence with the transformer backbone. We upsample the features back to the original sequence length with the U-Net decoder for the final prediction. We begin with the 2D U-Net design used by Nichol & Dhariwal (2021) for image diffusion and replace its 2D convolutions with corresponding 1D convolutions. For instance, we substitute each 2D convolution of size 3x3 with a 1D convolution of size 3. We make similar substitutions for the downsampling and upsampling operations. These changes enable the U-Net to handle 1D sequences. We provide complete details of our U-Net architecture in the appendix.

Transformers are naturally suited for handling 1D sequences. We utilize a pre-normalization transformer (Vaswani et al., 2017; Xiong et al., 2020) with RMSNorm (Zhang & Sennrich, 2019) and GeGLU activations (Shazeer, 2020). Because relative positional encodings tend to be more effective than absolute positional encodings (Shaw et al., 2018; Gulati et al., 2020), we encode positional

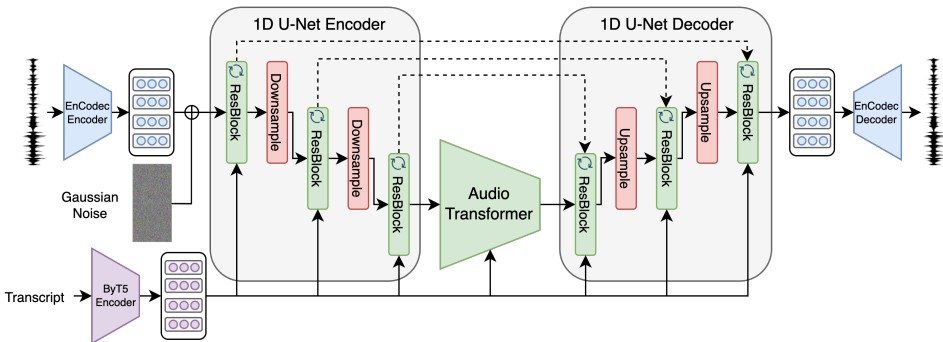

Figure 1: **Overview of Simple-TTS**. We learn our diffusion model in the latent space of the En-Codec audio autoencoder. We downsample the audio features with a 1D U-Net encoder, process it with a transformer, and upsample it with a 1D U-Net decoder for the final prediction. We condition our network on representations from a frozen ByT5 encoder.

information with a Dynamic Position Bias (DPB) (Wang et al., 2021; Liu et al., 2022). This intro-duces a lightweight MLP at the self-attention layers that maps relative offsets between locations, $\Delta x_{i,j} \in \{..., -1, 0, 1, 2, ...\}$, to head-specific bias terms that are added to the self-attention logits before the softmax.

To condition the network on the level of noise, we utilize $\alpha$-conditioning (Chen et al., 2021; Saharia et al., 2022b; Chen, 2023). We map $\alpha_t$ to a sinusoidal position embedding (Vaswani et al., 2017) and pass it through an MLP to obtain a time embedding. We follow standard practice and utilize adaptive group normalization layers in the residual blocks of the U-Net conditioned on the time embedding (Ho et al., 2020). For the transformer model, we similarly apply adaptive RMSNorm before the feedforward layers. We condition on textual representations extracted from a frozen, pre-trained language model, ByT5-Large. We mean-pool the representations from the ByT5 encoder and pass it through an MLP to generate a global text embedding that is added to the time embedding. We also introduce cross-attention layers to the transformer model that cross attend to the text representations.

We let the diffusion model determine the length of the speech during generation. We pad the audio with variable-length silence up to a maximum length of 20 seconds during training. To effectively reconstruct the audio representation at high noise levels, the denoising network must therefore learn to estimate the duration of the speech from the transcript and the optional speaker prompt. Dur-ing generation, the diffusion model then terminates the speech by generating silence which can be cheaply trimmed with an audio utility such as Sox. If a duration prediction model is available, it can likely be incorporated to accelerate generation, but it is not required for generation.

**Classifier-Free Guidance.** To enable the application of classifier-free guidance (Ho & Salimans, 2021), we drop the text with probability $p = 0.1$ and therefore jointly train a conditional and uncon-ditional diffusion model. During inference, we introduce a sampling parameter $w$, and compute

$$\hat{\mathbf{x}}_\theta^w(\mathbf{z}_t, t, \mathbf{c}) = \hat{\mathbf{x}}_\theta(\mathbf{z}_t, t) + w * (\hat{\mathbf{x}}_\theta(\mathbf{z}_t, t, \mathbf{c}) - \hat{\mathbf{x}}_\theta(\mathbf{z}_t, t)).$$

When $w = 1.0$, this reduces to the conditional diffusion model, and setting $w > 1.0$ increases the influence of the conditioning information. This technique enables us to trade off fidelity with sample diversity and is widely used by text-to-image diffusion models (Saharia et al., 2022a). For the cross-attention layers, we concatenate a learnable null embedding with the text features along the sequence dimension. We drop the conditioning information by masking out the text embeddings from the cross-attention mechanism and zeroing the mean-pooled text embedding. We set $w = 5.0$ by default, and examine the impact of guidance in our ablation studies.

**Speaker-Prompted Generation.** Diffusion models can perform speaker-prompted generation by treating it as an inpainting problem (Lugmayr et al., 2022; Le et al., 2023). We train our denoising network for both text-only and speaker-prompted TTS synthesis in a multi-task fashion. With prob-ability $p = 0.5$ we train the network to perform audio inpainting by concatenating a clean audio latent with a noisy latent vector. We sample a duration $d$ and concatenate the start of the latent audio representation $\mathbf{x}[:d]$ with the end of the noisy latent $\mathbf{z}_t[d:]$ to construct the input. We also

introduce a binary embedding to identify corrupted frames, which we sum with the input after the initial projection. When calculating the loss, we mask out frames corresponding to the clean audio.

For the prompt duration, we sample the proportion of the input, $d \in [0, 1]$, to hold out as the clean prompt. For instance, if we sample $d = 0.1$ for a 10 second clip of audio, then we use the frames corresponding to the first second of audio as the clean prompt. We utilize a Beta distribution with a mode of $.01$ and a concentration of $5$ as our sampling distribution, visualized in Figure 4 in the appendix. We chose this parameterization because it is bounded on the unit interval and has a strong leftward skew towards challenging settings with short prompts. Reasonable alternative distributions would likely be similarly effective. During inference, we preprend some sample audio and the associated text to the input to generate speech that is consistent with the provided sample.

**Diffusion Noise Schedule.** The diffusion noise schedule $\alpha_t$ influences the weighting placed on different levels of noise during training and is a critical factor in downstream sample quality. The cosine noise schedule, $\alpha_t = \cos{(.5\pi t)}$, introduced by Nichol & Dhariwal (2021), has become a common choice across applications and domains (Ho et al., 2022a; Saharia et al., 2022a; Janner et al., 2022; Chen et al., 2022). However, Hoogeboom et al. (2023) and Chen (2023) found that common noise schedules, such as the cosine schedule, are implicitly tuned for low-resolution images. Modifying the noise schedule to emphasize different levels of noise enables replacing pipelined approaches for high-resolution image diffusion with a single diffusion model. We should therefore not expect, a priori, that common image noise schedules will be effective for other modalities.

Both Hoogeboom et al. (2023) and Chen (2023) shift an existing noise schedule by some scale factor, $s$, to emphasize training at higher levels of noise. Given a noise schedule $\alpha_t$ with SNR $\lambda_t = \frac{\alpha_t^2}{1 - \alpha_t^2}$, the shifted noise schedule, $\alpha_{t,s} \in [0, 1]$, is defined such that

$$\frac{\alpha_{t,s}^2}{1 - \alpha_{t,s}^2} = \lambda_{t,s} = \lambda_t * s^2 = \frac{\alpha_t^2}{1 - \alpha_t^2} * s^2.$$

Given $\alpha_t$ and the scale factor $s$, the new noise schedule $\alpha_{t,s}$ has a closed-form solution. Using the fact that $\alpha_t^2 = \mathrm{sigmoid}(\log(\lambda_t))$ (see Kingma et al. (2021)), the shifted noise schedule can be computed in log-space (for numerical stability) as

$$\alpha_{t,s}^2 = \mathrm{sigmoid}(\log(\lambda_{t,s})) = \mathrm{sigmoid}(\log(\lambda_t * s^2)) = \mathrm{sigmoid}(\log(\lambda_t) + 2\log(s)).$$

To understand the effect of scaling the noise schedule, we can examine the WER of a pre-trained ASR model[1] on resynthesized audio for re-scaled latents,

$$\frac{\mathbf{z}_t}{\alpha_t} = \frac{(\alpha_t \mathbf{x} + \sqrt{1 - \alpha_t^2}\epsilon)}{\alpha_t}, \epsilon \sim \mathcal{N}(\mathbf{0}, \mathbf{I}),$$

across time for different noise schedules. The re-scaling ensures that $\mathbb{E}[\mathbf{z}_t/\alpha_t] = \mathbb{E}[\mathbf{x}]$ while maintaining the SNR imposed by the noise schedule, $\mathrm{SNR}(\frac{\mathbf{z}_t}{\alpha_t}) = \mathrm{SNR}(\mathbf{z}_t) = \lambda_t$.

We visualize shifted cosine noise schedules with different scale factors and plot the WER across time for the LibriSpeech test-clean set in Figure 2. We observe that using a scale factor $s < 1$ emphasizes training at higher levels of noise. The intelligibility of the speech degrades with increasing levels of noise, and the scale factor controls the rate of degradation. When using the standard cosine noise schedule, the WER is nearly unaffected for $t \in [0, .3]$. Because we sample $t \in \mathcal{U}(0, 1)$ during training, this means that a third of training is dedicated to reconstructing highly intelligible speech.

Decreasing the scale factor increases the amount of training time spent at high noise levels where the words are being resolved and the transcript must be used to estimate the original data. This dedicates more diffusion steps to resolving the global structure of the speech, such as duration and word placement, compared to the standard noise schedule. We employ a shifted cosine noise schedule with a scale factor of 0.5 and this choice in our ablation studies.

## 5 EXPERIMENTS

**Datasets.** We utilize the English subset of the Multilingual LibriSpeech (MLS) dataset, which consists of 44.5K hours of speech derived from audiobooks from LibriVox (Pratap et al., 2020), to

---

[1]https://huggingface.co/facebook/hubert-large-ls960-ft

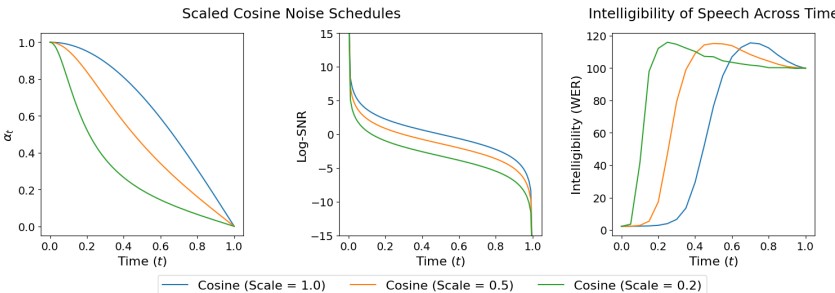

Figure 2: **Diffusion noise schedules.** In the left two plots, we visualize the cosine noise schedule with different scale factors. In the right plot, we visualize the intelligibility of the speech across time for different schedules. Reducing the scale factor allocates more time to higher levels of noise and accelerates the degradation of the speech.

train Simple-TTS. This dataset consists of audio from nearly 5,500 speakers which makes it well-suited for developing TTS models capable of synthesizing diverse voices. For evaluation, we utilize the widely studied LibriSpeech (LS) dataset (Panayotov et al., 2015). To enable direct comparison with prior work (Borsos et al., 2023; Wang et al., 2023; Le et al., 2023), we consider a filtered subset of LS test-clean consisting of clips between four and ten seconds in length.

**Model Implementation.** Our U-Net encoder and decoder each have 4 stages which downsample the input from 1504 frames to 188 frames at the lowest resolution. We utilize a feature dimensionality of 512 which is increased to 768 for the lowest resolution feature map. The transformer backbone has 12 layers and a feature dimension of 768. Simple-TTS has 243M trainable parameters, making it smaller than recent generative speech models such as Vall-E (302M[2]) and VoiceBox (364M). We train Simple-TTS for 200k steps with a batch size of 256 audio clips. We observe that our model is still improving at the end of training and additional training would likely be beneficial.

**Baselines.** For text-only synthesis, we compare against VITS (Kim et al., 2021), a variational autoencoder with adversarial training. We consider both VITS variants released by Kim et al. (2021): the single-speaker VITS-LJ trained on LJ Speech, and the multi-speaker VITS-VCTK trained on VCTK. We also compare against English MMS-TTS (Pratap et al., 2023), a recent single-speaker model utilizing the VITS architecture. For speaker-prompted TTS, we compare against YourTTS (Casanova et al., 2022), a VITS model conditioned on a speech prompt. Since other recent state-of-the-art generative models like Vall-E (Wang et al., 2023) and VoiceBox (Le et al., 2023) are not publicly available, we follow their evaluation protocols and compare against their reported metrics.

**Evaluation Metrics.** To evaluate the *intelligibility* of the synthesized audio, we transcribe the speech with a pre-trained ASR model and compute the WER between the transcribed text and original transcript. We use the same HuBERT-L model (Hsu et al., 2021) employed by prior work (Borsos et al., 2023; Wang et al., 2023; Le et al., 2023) to enable direct comparison[3]. For speaker-prompted TTS, we evaluate the *similarity* between the prompt and synthesized speech by utilizing the pre-trained speaker verification model employed by prior work[4] (Wang et al., 2023; Le et al., 2023). We follow the procedure of Wang et al. (2023) and extract speaker embeddings for the re-synthesized prompt and synthesized speech and report the cosine similarity between the embeddings.

We also conduct a human study to collect the mean opinion score (MOS) as a subjective evaluation of audio quality in the speaker-prompted setting. We follow the same setup as Wang et al. (2023) and select one synthesized utterance per speaker in the LS test-clean subset, leaving us with 40 audio samples. We recruit 11 human annotators and collect at least 10 annotations per sample. Subjective evaluations are collected on a 5-point scale, with 1 being the worst and 5 being the best. We follow Le et al. (2023) and collect a subjective measure of audio quality, the quality MOS (QMOS), and

---

[2]This value is estimated from the reported transformer hyperparameters.

[3]https://huggingface.co/facebook/hubert-large-ls960-ft

[4]The WavLM-Large model released at https://github.com/microsoft/UniSpeech/tree/main/downstreams/speaker_verification.

| Method | End-to-End | LibriSpeech Test-Clean | |
|---|---|---|---|
| | | Intelligibility (WER) ↓ | Similarity ↑ |
| Reference Human Speech | — | 2.2 | .754 |
| *Speech-to-Speech* | | | |
| GSLM (Lakhotia et al., 2021) | ✗ | 12.4 | .126 |
| AudioLM (Borsos et al., 2023) | ✗ | 6.0 | — |
| *Text-Only TTS* | | | |
| VITS-LJ (Kim et al., 2021) | ✓ | 4.2 | n/a |
| VITS-VCTK (Kim et al., 2021) | ✓ | 9.1 | n/a |
| MMS-TTS (Pratap et al., 2023) | ✓ | 7.2 | n/a |
| Simple-TTS (Ours) | ✓ | 2.4 | n/a |
| *Speaker-Prompted TTS* | | | |
| Vall-E (Wang et al., 2023) | ✗ | 5.9 | .580 |
| VoiceBox (Le et al., 2023) | ✗ | 1.9 | .681 |
| YourTTS (Casanova et al., 2022) | ✓ | 7.7 | .337 |
| Simple-TTS (Ours) | ✓ | 3.4 | .514 |

Table 2: **Automated Evaluation of TTS Systems.** Systems in gray are not publicly available.

a subjective measure of the similarity between the prompt and synthesized speech, the similarity MOS (SMOS). These metrics are not reproducible and cannot be directly compared across studies. We therefore collect judgements for the strongest open-source system, YourTTS, for comparison.

## 6    RESULTS

Our results in Table 2 demonstrate that our method can generate high-fidelity, intelligible speech in a text-only setting, nearly matching the word error rate of the ground truth audio. Notably, our text-only WER surpasses that of the single-speaker VITS-LJ and MMS-TTS models, while providing the additional capability of multi-speaker synthesis. When provided a three second speaker prompt, our model effectively generates high quality speech and maintains the characteristics of the prompt, with lower a WER than the state-of-the-art autoregressive model, Vall-E, and YourTTS. Additionally, for speaker similarity, we outperform the strongest open-source baseline, YourTTS, by a wide margin.

We report the results of our human study in Table 3. Simple-TTS synthesizes higher quality audio that is more similar to the prompt than YourTTS. It achieves statistically significant improvements on both quality MOS (QMOS) and similarity MOS (SMOS), with gains of +0.52 and +1.46 points respectively. The human study validates that Simple-TTS produces speech of higher quality and greater prompt similarity compared to the state-of-the-art publicly available TTS system.

**Sampling Configurations.** We examine the importance of classifier-free guidance in Figure 3 when using both the stochastic DDPM sampler and the deterministic DDIM sampler (Song et al., 2020a). Similar to text-to-image diffusion models, classifier-free guidance is critical for generating speech that is faithful to the provided transcript. For both settings, the conditional diffusion model ($w = 1.0$) exhibits poor intelligibility, but improves rapidly with guidance, outperforming the autoregressive Vall-E when $w = 1.5$ in the speaker-prompted setting. The speaker similarity also improves with stronger guidance, demonstrating that guidance also improves the quality of the speech. Both the DDPM and DDIM samplers lead to intelligible speech that is well-aligned with the prompt. For image diffusion models, Karras et al. (2022) observed that stochastic sampling effectively corrects errors made in earlier sampling steps, but leads to a gradual loss of detail. In our setting, we find that the DDPM sampler corrects errors that impact intelligibility, leading to consistently lower WERs, while the deterministic DDIM better preserves fine details concerning the speaker identity.

We present the performance across different numbers of sampling timesteps in Figure 3. In the text-only setting, we surpass the intelligibility of VITS-LJ using just 15 sampling steps with a WER of 4.2. In the speaker-prompted setting, we achieve a WER of 4.0 with 15 sampling steps, surpassing

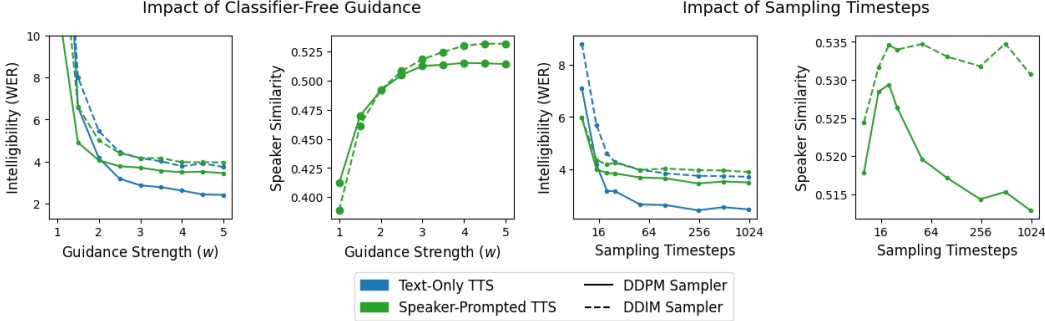

Figure 3: **Exploration of sampling configurations.** In the left two plots, we report the impact of varying the strength of classifier-free guidance. In the right two plots, we report the effect of using different numbers of sampling steps.

| Source | QMOS | SMOS |
|--------|------|------|
| Reference | $4.20_{(4.11, 4.28)}$ | $3.87_{(3.76, 3.98)}$ |
| YourTTS | $2.05_{(1.96, 2.14)}$ | $1.56_{(1.49, 1.64)}$ |
| Simple-TTS | $\mathbf{2.77}_{(2.69, 2.85)}$ | $\mathbf{3.04}_{(2.94, 3.14)}$ |

Table 3: **Subjective human evaluation.** We report mean and 95% confidence intervals from bootstrapping.

| Text Encoder | Noise Schedule | WER $\downarrow$ |
|--------------|----------------|------|
| ByT5-Large | Cosine (s=0.5) | 6.0 |
| T5-Large | Cosine (s=0.5) | 26.6 |
| ByT5-Large | Cosine (s=1.0) | 13.0 |
| ByT5-Large | Cosine (s=0.2) | 9.6 |

Table 4: **Ablation studies.** Models are trained for 50k steps.

the state-of-the-art autoregressive model, Vall-E. Increasing the sampling steps generally improves quality, but the stochasticity gradually degrades speaker similarity with the DDPM sampler.

**Ablation Studies.** We ablate our choice of pre-trained language model and noise schedule in Table 4. For our ablation studies, we train each model for 50k steps and report the WER for text-only TTS to quantify the speech-text alignment. We observe that utilizing the T5-Large encoder instead of the ByT5-Large encoder dramatically increases the error rate, by a factor of $4.4\times$. This demonstrates the importance of using character-aware language representations instead of BPE encodings.

Re-scaling the cosine noise schedule is also essential for strong alignment. Training our model with the standard cosine noise schedule increases the error rate by a factor of $2.2\times$ compared to our noise schedule. We also ablate a more aggressive scale factor of s=0.2 and observe a degradation in performance compared to s=0.5, although it still outperforms the standard noise schedule. These ablation studies demonstrate the critical importance of both the character-aware language representations from ByT5 and of dedicating more time to noise levels where the global structure of the speech is being resolved with the scaled diffusion noise schedule.

## 7    CONCLUSION

In this work, we present Simple-TTS, the first diffusion model capable of end-to-end text-to-speech synthesis. We demonstrate that latent diffusion is an effective approach for speech generation without requiring explicit alignment or duration modeling. The key ingredients in the success of Simple-TTS are: adapting a vision diffusion backbone for efficient sequence modeling of audio, utilizing a byte-level language model to capture linguistic properties critical for natural speech synthesis, and modifying the diffusion noise schedule to improve text-speech alignment. Together, these innovations enable Simple-TTS to perform speech synthesis directly from text, without external alignment tools, pipelines, or extra components. Simple-TTS generates natural, intelligible speech, nearing human-level word error rates. We are excited about applications in controllable speech synthesis and further advancements enabled by end-to-end generative modeling of speech.

## 8 REPRODUCIBILITY STATEMENT

We conduct this work on publicly available datasets and utilize publicly available models for the audio autoencoder and pre-trained language model. We outline full implementation details relating to the model architecture, training setup, and hyperparameters in Appendix C. We will open-source our implementation upon acceptance and will release the Simple-TTS checkpoint to the community.

## 9 ETHICS STATEMENT

An effective speech generation model has many positive applications such as improving the quality of vocal assistants and accessibility tools. However, like any generative modeling technique, it carries risks if used maliciously. Our system should not be utilized for the unauthorized synthesis or impersonation of others' voices without consent. We plan to release our model as a valuable resource to the community, but urge users to employ it responsibly. We also encourage further research into the development of methods for automated detection of synthesized or manipulated speech.

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

## A    QUALITATIVE EXAMPLES

We provide uncurated qualitative examples for Simple-TTS in both the text-only and speaker-prompted setting in the supplemental materials. For reference, we also provide synthesized speech with YourTTS for the same speech prompts.

## B    ADDITIONAL FIGURES

We visualize the distribution used for sampling prompt durations for speaker-prompted inpainting in Figure 4.

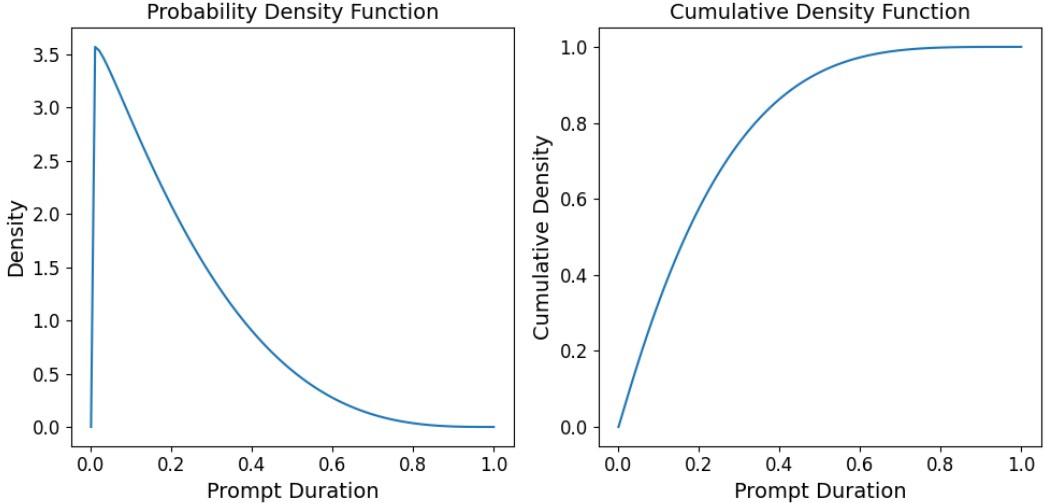

Figure 4: **Sampling Distribution of Prompt Durations.** We utilize a Beta distribution with a mode of .01 and a concentration of 5 to sample relative prompt durations during training to enable speaker-prompted inpainting. We selected this distribution to concentrate mass around short durations. The x-axis shows the prompt duration as a fraction of the full audio length. We sample a relative duration from this distribution and multiply it by the length of the audio clip to obtain the prompt duration in seconds.

## C    IMPLEMENTATION DETAILS

We report hyperparameters for Simple-TTS in Table 5.

**U-Net Implementation.** Our 1-D U-Net is modeled after the 2-D U-Net used by Nichol & Dhariwal (2021) for image diffusion. Each stage in the U-Net has two residual blocks. The residual block consists of a sequence of two pre-activation (He et al., 2016) 1D convolutions with a width of 3. We downsample the sequence length using a 1D convolution with a width of 4 and a stride of 2. To upsample the sequence length, we perform nearest neighbor upsampling followed by a 1D convolution of width 3. We also apply a lightweight linear attention mechanism (Shen et al., 2018; Katharopoulos et al., 2020) before the downsampling and upsampling operations to model global feature interactions at high resolution stages. We condition the U-Net on the timestep representation using adaptive group normalization and also follow the recommendation of Peebles & Xie (2022) and apply a time-conditioned gate to the outputs of the residual connections. We scale the skip connections in the U-Net by $\frac{1}{\sqrt{2}}$, following Song et al. (2020b); Saharia et al. (2022b).

**Transformer Implementation.** We utilize a standard pre-normalization transformer (Vaswani et al., 2017; Xiong et al., 2020) with RMSNorm (Zhang & Sennrich, 2019) and GeGLU activations (Shazeer, 2020). We condition the transformer on the timestep by applying adaptive RMSNorm

before the feedforward layers with time-dependent residual gating (Peebles & Xie, 2022). We utilized query-key RMSNorm (Dehghani et al., 2023) for the cross-attention mechanism because we observed that it improved stability in preliminary experiments. We did not ablate this decision carefully for our final model, so this design decision may not be necessary.

**Sampling Configuration.** We explore both the stochastic DDPM sampler and the deterministic DDIM sampler. For the stochastic DDPM sampler, the upper and lower bounds on the variance for the reverse process from time $t$ to time $s$, where $0 \leq s < t \leq 1$, are given by $\sigma_{\max}^2 = 1 - \alpha_{t|s}^2$ and $\sigma_{\min}^2 = \frac{1-\alpha_s^2}{1-\alpha_t^2} * (1 - \alpha_{t|s}^2)$ respectively. We follow Hoogeboom et al. (2023) and set the variance for the DDPM sampler to a log-scale interpolation between the upper and lower bounds $\sigma^2 = \exp(v \log(\sigma_{\max}^2) + (1 - v) \log(\sigma_{\min}^2))$ with $v = 0.2$. We did not explore this choice in detail and further exploration may improve performance.

| Model Configuration | |
|---|---|
| *1D U-Net* | |
| Audio Sequence Length | 1504 |
| EnCodec Feature Dimension | 128 |
| U-Net Feature Dimension | 512 |
| Residual Blocks Per Stage | [2, 2, 2, 2] |
| Channel Multipliers | [1, 1, 1, 1.5] |
| Linear Attention Head Dim | 32 |
| Number of Linear Attention Heads | 4 |
| *Transformer* | |
| Sequence Length | 188 |
| Feature Dimension | 768 |
| Feedforward Dimension | 2048 |
| Number of Layers | 12 |
| Attention Head Dim | 64 |
| Number of Attention Heads | 12 |
| Positional Encoding | Dynamic Position Bias (Wang et al., 2021; Liu et al., 2022) |
| Training Configuration | |
| Optimizer | AdamW (Loshchilov & Hutter, 2019) |
| Adam $\beta_1$, $\beta_2$ | 0.9, 0.999 |
| Learning Rate | 1e-4 |
| Weight Decay | 0.0 |
| Diffusion Regression Loss | L1 |
| Warmup Steps | 1000 |
| EMA Decay | 0.9999 |
| Learning Rate Schedule | Cosine Decay |
| Gradient Clipping | 1.0 |
| Global Batch Size | 256 |
| Per-Device Batch Size | 16 |
| Gradient Accumulation Steps | 2 |
| GPUs | 8x Nvidia A10G |
| Precision | bfloat16 |
| Training Steps | 200k |

Table 5: Implementation details for Simple-TTS.

# D   HUMAN STUDY INSTRUCTIONS

We present the instructions used in our human study.

## D.1   QMOS

**Task Instructions.** In this task you will hear samples of speech recordings. The purpose of this test is to evaluate the quality and intelligibility of each file in terms of its overall sound quality and

the amount of mumbling and unclear phrases in the recording. Please keep in mind that speech samples can be distorted and noisy, however these are only specific examples. Please use a headset for listening and adjust your volume level to your comfort during this training, and do not change later during the experiment. You should give a score according to the following scale, known as the MOS (mean opinion score) scales:

**Score (Quality and Intelligibility of the speech):**   1 (Bad) 2 (Poor) 3 (Fair) 4 (Good) 5 (Excellent)

### D.2   SMOS

**Task Instructions.**   Your task is to evaluate the similarity of the synthesized speech samples to the given speech prompt. You should focus on the similarity of the speaker, speaking style, acoustic conditions, background noise, etc. You should rank the recordings on the scale between 1-5, where 5 is the best quality and 1 is the worst. In other words, please rank the recordings according to their acoustic similarity to the given prompt, meaning as if they were recorded in the same place by the same speaker speaking in similar styles. This task typically requires approximately 120 seconds to complete. Please use a headset for listening and adjust your volume level to your comfort during this training, and do not change later during the experiment.

**Score:**   1 (Bad) 2 (Poor) 3 (Fair) 4 (Good) 5 (Excellent)

