# OpenReview forum: "Simple-TTS: End-to-End Text-to-Speech Synthesis with Latent Diffusion"
_ICLR.cc/2024/Conference — Submitted to ICLR 2024_

### Official Review · Reviewer_nfMt · 2023-10-28

**Soundness:** 1 poor
**Presentation:** 1 poor
**Contribution:** 1 poor
**Rating:** 1
**Confidence:** 5

**Summary:**

They proposed a text-to-speech model which utilizes a latent diffusion model. They introduce a U-ViT for latent diffusion based speech synthesis, and they do not require text-speech alignment for speech synthesis.

**Strengths:**

This work utilizes a pre-trained language model for text encoder, and they can generate a speech without text and speech alignment. This facilitates training pipeline efficiently.

**Weaknesses:**

Although this work proposed a simple method for text-to-speech without text and speech alignment, I have many questions about this and others.

1. They fixed the maximum length of speech by 20 seconds during training. This may make a training pipeline simple, but I think it is not efficient for GPU. In addition, this framework could not control the duration of speech.

2. It would be better if you could add an additional experiment according to text length. Because you train your model with a fixed length, you should demonstrate the robustness according to text length.

3. The authors may not know the definition of end-to-end. This model is not the end-to-end speech synthesis model. They need the pre-trained audio autoencoder and language model for the text encoder. They have overclaimed it.

4. The audio quality is too bad. I think it is because the audio autoencoder has a lower quality. You should have trained an audio autoencoder with speech data or replaced it with a high-quality audio autoencoder. I recommend to use different audio autoencoder such as DAC or HiFi-Codec or utilizes a pre-trained codec-based vocoder such as Vocos for high-quality waveform audio. It is well known that re-training the codec-based vocoder could generate a better quality of audio.

5. I think NaturalSpeech 2 already introduced this kind of method for speech synthesis. The difference with NaturalSpeech 2 is only the necessity of duration predictor. However, I think removing the duration predictor decreased the controllability of the model so I hope the authors address this issue. I think that the authors should have listened to the demo samples of NaturalSpeech 2 and compared the audio quality with yours. In addition, although they propose new architecture, there is no ablation study for model architecture.

6. They only compared the model with YourTTS. YourTTS has a very low audio quality. You should have trained the VITS with same dataset and speaker prompt.

**Questions:**

I sincerely have a question about the audio quality. How do the authors think the audio quality of your model.

---

### Official Review · Reviewer_Smvk · 2023-10-31

**Soundness:** 2 fair
**Presentation:** 2 fair
**Contribution:** 2 fair
**Rating:** 3
**Confidence:** 3

**Summary:**

This work proposes a TTS model, the simple-TTS, that uses latent diffusion models and U-Audio Transformer.

**Strengths:**

1. The usage of U-Audio Transformer is potentially useful for speech generation.
2. The results reported in the experimental section are encouraging.

**Weaknesses:**

1. The authors claim that the proposed model is an end-to-end model, however, it contains at least 3 separately training stages. Obviously, it is a not end-to-end model.

2. The authors call this model is a 'simple' model, however, it's not so simple. Taking the baseline model YourTTS for comparison, YourTTS is trained without using language model pre-training and En-Codec model, thus appears to be more simple.

3. In sections 1 and 2, the authors claim that simple-TTS is much simpler than NaturalSpeech2 and VoiceBox. However, in the experimental section, they do not provide a direct comparison to these models.

4. The experiments are far from sufficient.
   1). Why not present the MOS results for Text-only TTS?  which is very important to evaluate this work.
   2). Many important details are missing. For example, the sample rate of the audio samples. The synthesis speed is not presented.
   3). From my subjective evaluation, the audios in the supplementary Material are not as good as some SOTA models such as VITS and NatrualSpeech.

**Questions:**

Why not present the MOS results for Text-only TTS?

---

### Official Review · Reviewer_etZW · 2023-11-01

**Soundness:** 3 good
**Presentation:** 3 good
**Contribution:** 3 good
**Rating:** 5
**Confidence:** 5

**Summary:**

This paper proposes a TTS model in the form of Latent diffusion called Simple-TTS. It simplifies the training process of the TTS model by using a pre-trained text encoder and EnCodec and training only the weight of latent diffusion model. It outperforms the open-source zero-shot TTS model, YourTTS.

**Strengths:**

1. As the title of the paper, the training process of TTS has been greatly simplified. By utilizing a pre-trained Text Encoder (ByT5), the need to train the text encoder has been eliminated, and by aligning speech with text through cross-attention, the need for a duration predictor has been removed. Through this, the model is trained using only a simple v-prediction for diffusion model.

2. Listening to the samples provided in the Supplementary material, the generated samples sound expressive.

**Weaknesses:**

1. In simplifying the model training, there is a suspicion of potential issues in the process of learning monotonic alignment between text and speech through cross attention. Additionally, padding all sentences to a fixed length during training and allowing the diffusion model to learn on its own is presumed to be heavily influenced by the length of the speech data in the dataset. It seems that this model may also have the robustness issues that were present in autoregressive TTS models using cross-attention for alignment like Tacotron or TransformerTTS.

2. Sample quality in supplementary material is too bad.

3. Despite the proposed model has lower speaker adaptation performance compared to recent papers such as VALL-E, SPEAR-TTS, and VoiceBox, claiming to release the strongest publicly available system by showing performance improvements over an easily beatable baseline like YourTTS seems like an overstatement in abstract.

**Questions:**

* Despite using pre-trained models, why do you refer to Simple-TTS as an end-to-end TTS model?

* Regarding Weakness 1, does Simple-TTS have no robust issues in finding alignments even with the introduction of cross-attention? If not, it would be beneficial to provide the ASR metrics for the Hard sentences found in Appendix B of the FastSpeech paper as well.

* How can Simple-TTS generate speech that exceeds the predetermined length during training? For example, generating 30+ seconds of speech given a few long sentences.

* Regarding Weakness 2, the sample quality is too bad compared to existing zero-shot TTS models. NaturalSpeech 2 also models continuous latent representation similarly to the proposed paper, but the sample quality of the proposed method is relatively poor in comparison. It would be beneficial if this could be improved.

---

### Meta-Review · Area_Chair_KRS4 · 2023-12-04

**Metareview:**

The paper introduces a simplified TTS training process that leverages a pre-trained Text Encoder (ByT5), thereby eliminating the need for a separate text encoder and duration predictor. However, there are concerns regarding the poor quality of the samples provided in the supplementary material, the lack of comprehensibility in the experimental validation, and not a real end-to-end model. All three reviewers consistently rated the paper below the acceptance threshold.

**Justification For Why Not Higher Score:**

N/A

**Justification For Why Not Lower Score:**

N/A

---

### Decision · Program_Chairs · 2024-01-16

Reject